# VWCE modulates amino acid-dependent mTOR signaling and coordinates with KICS-TOR to recruit GATOR1 to the lysosomes

Tianyu Zhao[1,2], Yuanyuan Guan[1,2], Chenchen Xu[1,2], Dong Wang[1,2], Jialiang Guan[3] & Ying Liu ◉ [1,2,4] ✉

The mechanistic target of rapamycin complex 1 (mTORC1) is a crucial regulator of cell growth. It senses nutrient signals and adjusts cellular metabolism accordingly. Deregulation of mTORC1 has been associated with metabolic diseases, cancer, and aging. Amino acid signals are transduced to mTORC1 through sensor proteins and two protein complexes named GATOR1 and GATOR2. In this study, we identify VWCE (von Willebrand factor C and EGF domains) as a negative regulator of amino acid-dependent mTORC1 signaling. Knockdown of VWCE promotes mTORC1 activity even in the absence of amino acids. VWCE interacts with the KICSTOR complex to facilitate the recruitment of GATOR1 to the lysosomes. Bioinformatic analysis reveals that expression of VWCE is reduced in prostate cancer. More importantly, overexpression of VWCE inhibits the development of prostate cancer. Therefore, VWCE may serve as a potential therapeutic target for the treatment of prostate cancers.

The mechanistic target of rapamycin complex 1 (mTORC1) is a master regulator of cell growth that responds to a diverse set of environmental cues, including amino acids[1]. Activation of mTORC1 promotes anabolic processes such as protein synthesis, whereas suppression of mTORC1 activates catabolic pathways such as autophagy[1]. Therefore, the activity of mTORC1 must be precisely controlled[2–4].

Activation of mTORC1 is mediated by two types of GTPases: the Rheb (RAS homologue enriched in brain) GTPase that directly activates mTORC1 on the lysosomal surface, and the Rag GTPases that recruit mTORC1 to the lysosomes[5,6]. Mammals have four different Rags, which can be divided into two paralogous pairs, RagA/B and RagC/D. In response to amino acids, GTP-bound RagA/B and GDP-bound RagC/D form active heterodimers to recruit mTORC1 to the lysosome surface, the site of its activation[5,6]. Several protein complexes have also been identified to regulate amino acid-dependent mTORC1 activity. For example, GATOR1 (composed of DEPDC5, NPRL3, and NPRL2) is a GTPase-activating protein (GAP)

for RagA/B that negatively regulates mTORC1[7]. GATOR2 (composed of WDR59, WDR24, MIOS, SEH1L, and SEC13) positively regulates mTORC1 through unknown mechanisms to inhibit GATOR1[7]. In addition, KICSTOR (composed of KPTN, ITFG2, C12orf66, and SZT2) acts as the anchor site of GATOR1 on lysosomes and negatively regulates mTORC1[8,9]. However, the potential existence of yet unidentified regulators in the amino acid-induced activation of mTOR warrants deeper investigation to fully comprehend the intricacies of this pathway.

In this study, we utilize the interactome database to identify proteins capable of interacting with at least two established components of the mTORC1 pathway. Our findings pinpoint VWCE as a negative modulator of amino acid-dependent mTORC1 signaling. VWCE interacts with KICSTOR, which is instrumental in guiding GATOR1 to the lysosomal membrane. Bioinformatics analyses indicate a downregulation of VWCE in prostate cancer tissues. Importantly, overexpression of VWCE impairs prostate cancer progression.

[1]State Key Laboratory of Membrane Biology, New Cornerstone Science Laboratory, Institute of Molecular Medicine, College of Future Technology, Peking University, Beijing 100871, China. [2]Peking-Tsinghua Center for Life Sciences, Academy for Advanced Interdisciplinary Studies, Peking University, Beijing 100871, China. [3]PKU-Tsinghua-NIBS Graduate Program, Academy for Advanced Interdisciplinary Studies, Peking University, Beijing 100871, China. [4]Beijing Advanced Innovation Center for Genomics, Beijing 100871, China. ✉e-mail: ying.liu@pku.edu.cn

## Results

### Identification of VWCE as a negative regulator of mTORC1

To identify previously uncovered regulators that might affect amino acid-dependent mTORC1 activity, we employed interactome database BioPlex and BioGRID[10], and identified 45 proteins that can interact with at least two known components of the mTORC1 pathway (Supplementary Fig. 1a and Supplementary Table 1). We then used siRNA-mediated knockdown to test if any of these candidate proteins modulate the acute response to amino acid starvation or restimulation. HEK293T cells cultured with complete growth medium were starved of total amino acids for 50 min, and then restimulated with amino acids for 10 min (Fig. 1a). The kinase activity of mTORC1 was monitored by assessing the phosphorylation level of its direct substrate p70 S6 kinase 1 (S6K1)[11]. Notably, knockdown of VWCE (von Willebrand factor C and EGF domains) with siRNA or shRNA impaired the suppression of mTORC1 activity upon amino acid starvation (Fig. 1b and Supplementary Fig. 1b). The knockdown efficiency of VWCE was confirmed both at the mRNA level through real-time quantitative PCR (RT-qPCR) (Fig. 1c and Supplementary Fig. 1c), and at the protein level via flow cytometry analysis (Supplementary Fig. 1d, e, see Supplementary Fig. 2 for gating strategy) and immunoblotting (Supplementary Fig. 1f, g). VWCE possesses a predicted N-terminal signal sequence that likely targets it to the ER lumen during translation. It also manifests traits commonly associated with secreted proteins according to the UniProt database. Yet, some studies indicate VWCE's presence in the cytoplasm, functioning as a regulatory component in the β-catenin

signaling pathway[12]. This hints at the multifaceted roles of VWCE, suggesting it might either be secreted from cells or retained internally through mechanisms yet to be elucidated. To validate VWCE's involvement in mTORC1 regulation, we overexpressed VWCE in VWCE-deficient cells and observed a restored sensitivity to amino acid starvation in these cells. These findings exclude the possibility that the observed results of VWCE siRNA treatment were due to off-target effects, underscoring VWCE's role as an mTORC1 regulator (Fig. 1d). Moreover, VWCE-mediated mTORC1 regulation was not limited to a specific amino acid, as knockdown of VWCE compromised the ability of cells to inhibit mTORC1 activity when cells were starved of leucine, isoleucine, valine, arginine, or lysine (Supplementary Fig. 1h). In contrast, the sensitivity of mTORC1 to serum starvation was not impaired by VWCE deficiency (Supplementary Fig. 1i). These results indicate that VWCE plays a general role in modulating amino acid-regulated mTOR signaling. Previous studies showed that in cells restimulated with amino acids, mTOR translocates from the cytosol to the lysosomes, where it can be activated by the Rheb protein[5,6]. Consistent with the above results, knockdown of VWCE resulted in constitutive lysosomal localization of mTOR, even in the absence of amino acids (Fig. 1e, f).

### VWCE interacts with the KICSTOR complex

To investigate how VWCE regulates mTORC1 activity, we performed an epistatic analysis between VWCE and key components of the amino acid-sensing branch upstream of mTORC1. WDR59 deficiency inhibited amino acid-stimulated mTORC1 activity, while in cells deficient of

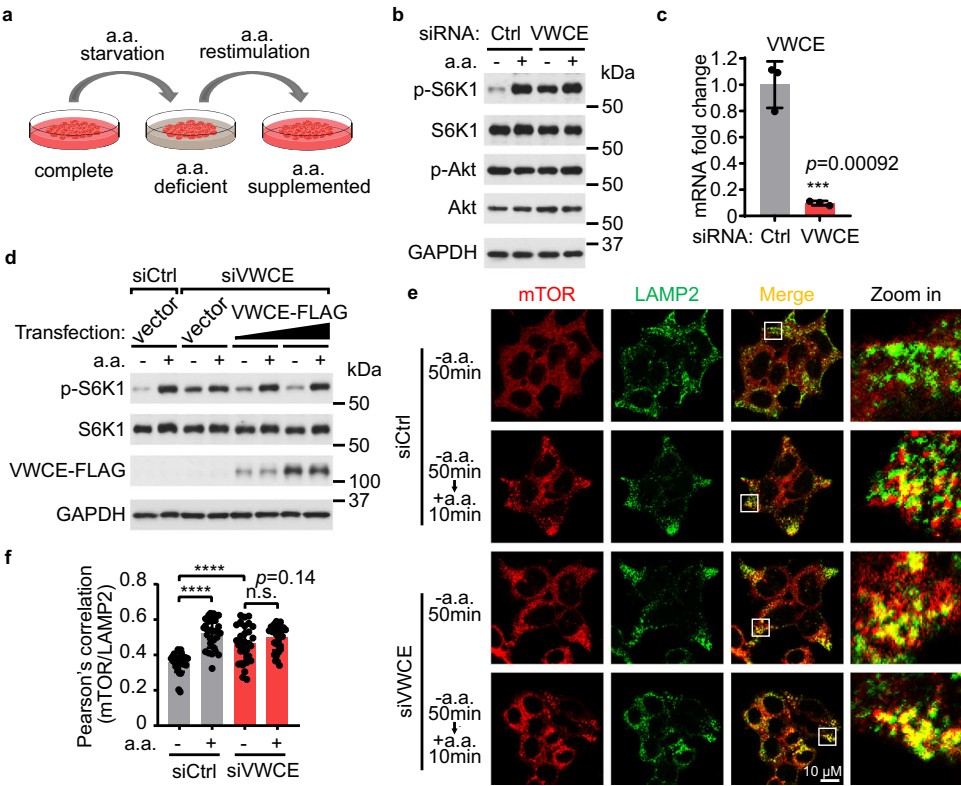

**Fig. 1 | VWCE regulates amino acid-dependent mTORC1 activity. a** A schematic diagram depicting amino acid (a.a.) deprivation and restimulation. **b** Knockdown of VWCE impairs the suppression of mTORC1 activity upon amino acid deprivation. Cells transfected with non-targeting siRNA (siCtrl) or siRNA targeting VWCE (siVWCE) were starved of amino acids for 50 min (-), or starved for 50 min and restimulated with amino acids for 10 min (+). Cell lysates were analyzed by immunoblotting. **c** The knockdown efficiency of VWCE in (**b**) is tested by RT-qPCR (*N* = 3). Data are mean ± s.d. ***p < 0.001 (unpaired two-sided Student's *t*-test). **d** Overexpression of VWCE restores the ability of VWCE-deficient cells to sense amino acid deprivation. The cells were transfected with the indicated siRNAs and

then transfected with the indicated plasmids after 6 h. After another 6 h, the cell culture medium was replaced with fresh medium, and the cells were allowed to grow for 60 h before harvesting. **e** mTOR (red) constitutively localizes on lysosomes in VWCE-knockdown cells, even in the absence of amino acids. The cells were treated by amino acid starvation or starvation plus restimulation before preparation for immunostaining. LAMP2 (green) was used as a lysosome marker. **f** Quantification of co-localization between mTOR and LAMP2 in (**e**) (*N* = 30). Data are mean ± s.d. ****p < 0.0001 (unpaired two-sided Student's *t*-test). The immunoblotting assays were independently replicated three times with consistent results. Source data are provided as Source data files.

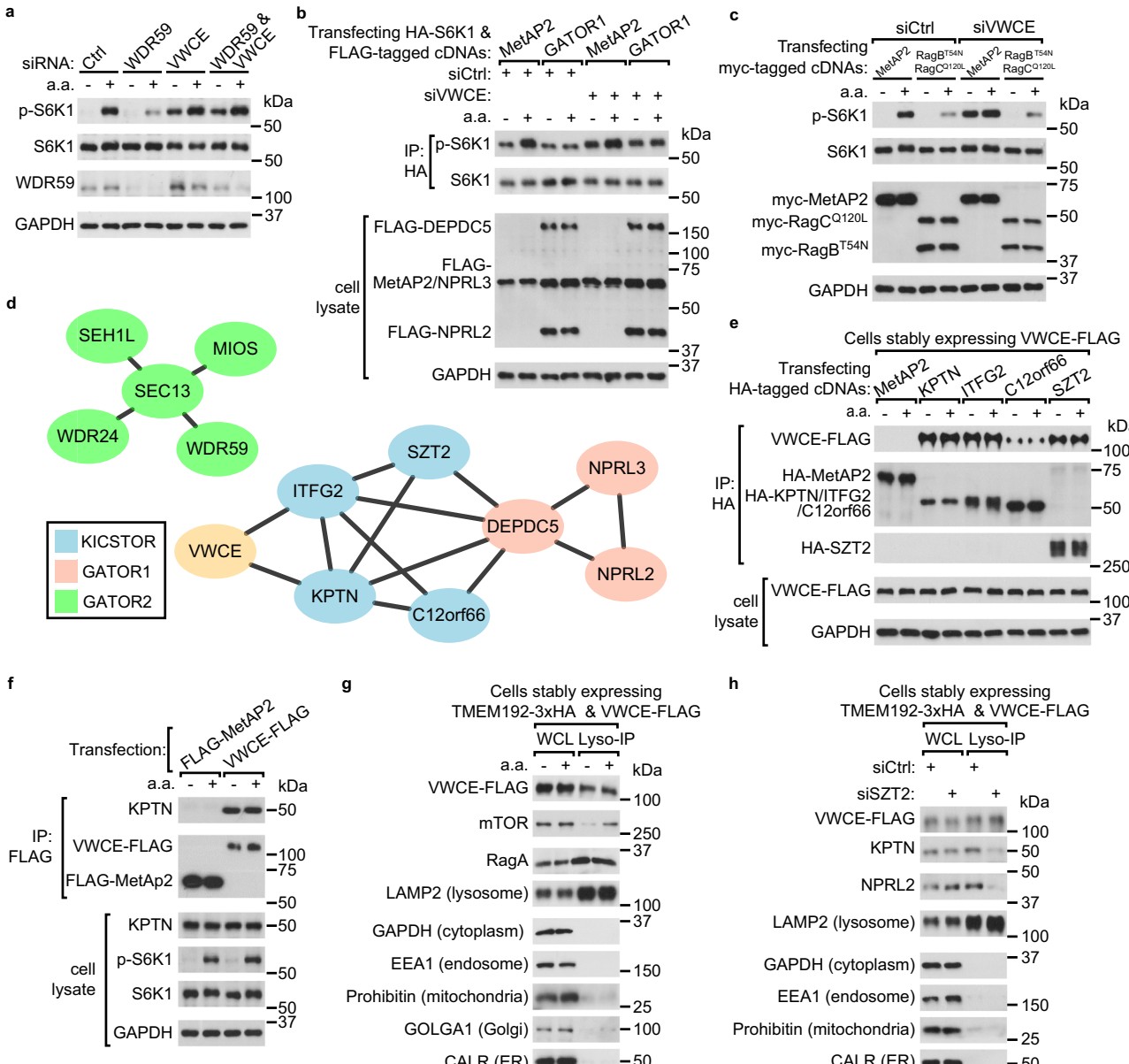

**Fig. 2 | VWCE interacts with KICSTOR and localizes on the lysosomes. a** VWCE acts downstream of GATOR2 to regulate mTORC1. Cells transfected with siCtrl, siWDR59, siVWCE, or siWDR59 plus siVWCE were treated by amino acid starvation (-) or starvation then restimulation (+) and lysed for immunoblotting. **b** GATOR1 overexpression inhibits mTORC1 activity to a lesser extent in VWCE-deficient cells. Cells transfected with the indicated siRNAs and plasmids were harvested for immunoprecipitation and immunoblotting. MetAP2 serves as a control. **c** VWCE acts upstream of Rag GTPases to regulate mTORC1. The siRNA-transfected cells were then transfected with dominant-negative Rag heterodimers (RagB[T54N]+RagC[Q120L]). **d** The interaction network of VWCE, KICSTOR components, GATOR1 components and GATOR2 components from BioPlex. **e** VWCE interacts with all 4 subunits of the KICSTOR complex (KPTN, ITFG2, C12orf66, SZT2). Cells stably expressing FLAG-tagged VWCE were transfected with the indicated HA-tagged cDNAs. Cell lysates and immunoprecipitates were analyzed by immunoblotting after the indicated amino acid treatments. **f** VWCE interacts with KPTN in an amino acid-independent manner. Cells transiently expressing the indicated cDNAs were lysed after the indicated amino acid treatments. Immunoprecipitates and lysates were analyzed by immunoblotting. **g** Amino acid-insensitive localization of VWCE on lysosomes. Lysosomes were immunopurified from HEK293T cells stably expressing 3×HA-tagged TMEM192 and FLAG-tagged VWCE. Purified lysosomes (Lyso-IP) and whole cell lysates (WCL) were analyzed by immunoblotting (**g, h**). Organelle-specific marker proteins are indicated. **h** Lysosomal localization of VWCE is independent of the KICSTOR complex. The experiments were independently replicated three times with consistent results. Source data are provided as a Source data file.

both WDR59 and VWCE, mTORC1 was constantly activated (Fig. 2a). This result indicates that VWCE acts downstream of GATOR2 to regulate mTORC1. Moreover, GATOR1 overexpression inhibited mTORC1 signaling to a lesser extent in VWCE-silenced cells compared with control cells (Fig. 2b), which suggests that VWCE is required for GATOR1 to inhibit mTORC1 efficiently. Lastly, expression of dominant-negative Rag mutants (RagB[T54N]+RagC[Q120L]) impaired mTORC1 activation in response to amino acid restimulation, regardless of the

presence or absence of VWCE (Fig. 2c). This indicates that VWCE functions upstream of Rag GTPases to regulate mTORC1.

To further elucidate the mechanism by which VWCE regulates mTORC1, we tried to find the interacting proteins of VWCE. The Bio-Plex database suggests that VWCE may interact with KPTN and ITFG2 (Fig. 2d), components of the KICSTOR complex[8]. Indeed, immunoprecipitation experiments demonstrated that VWCE associates with all KICSTOR components in an amino acid-insensitive manner (Fig. 2e, f

and Supplementary Fig. 3a, b). No interaction was observed between VWCE and GATOR1, GATOR2, or Rag (Supplementary Fig. 3c–e). In addition, when cells were overexpressed with VWCE, an elevated proportion of KPTN proteins were eluted in fractions with higher molecular weights, as determined by size-exclusion chromatography (SEC) analysis (Supplementary Fig. 3f). These results suggest that VWCE may regulate mTORC1 through KICSTOR. The VWCE-KICSTOR interactions were not disrupted by knockout of GATOR1 or GATOR2 (Supplementary Fig. 3g), which indicates that the VWCE-KICSTOR interactions were independent of the GATOR complexes.

Considering KICSTOR's role in guiding GATOR1[8] to lysosomes and the interaction between VWCE and KICSTOR, as well as VWCE's function downstream of GATOR2 in regulating mTORC1, we further explored the subcellular localization of VWCE. Immunostaining of endogenous FLAG-tagged VWCE revealed its presence in both the cytoplasm and nucleus. Within the cytosol, a portion of VWCE was observed to co-localize with lysosomes, and this localization remained consistent regardless of amino acid availability (Supplementary Fig. 3h, i). To further confirm the lysosomal localization of VWCE, we isolated lysosomes with high purity using an immuno-capture strategy ("LysoIP")[13] and observed an amino acid-insensitive localization of VWCE on the lysosomes (Fig. 2g and Supplementary Fig. 3j). The lysosomal VWCE could be digested by trypsin, suggesting that VWCE is located on the cytosolic surface of the lysosomes (Supplementary Fig. 3k, l). Unlike KPTN and NPRL2, the lysosomal localization of VWCE was not affected by knockdown of SZT2 (Fig. 2h). The knockdown efficiency of SZT was confirmed by the reduced mRNA level of SZT2 and the constitutive activation of mTORC1 even under amino acid starvation (Supplementary Fig. 3m, n). These results indicate that the lysosomal localization of VWCE is independent of the KICSTOR complex.

To discern which subunit mediates the VWCE-KICSTOR interaction, we generated knockout cell lines by depleting each KICSTOR component: KPTN, ITFG2, C12orf66, and SZT2 in HEK293T cells (Supplementary Fig. 4a, b). We observed that the interactions between VWCE-KPTN and VWCE-ITFG2 remained unaffected despite the deficiency of any other KICSTOR components (Supplementary Fig. 4c–e). Intriguingly, the VWCE-C12orf66 and VWCE-SZT2 interactions were compromised in the absence of either KPTN or ITFG2 (Supplementary Fig. 4f, g). Prior research has shown that the KPTN-ITFG2 interaction can persist independently of the full KICSTOR complex[8]. We also observed that knockout of ITFG2 affected the level of KPTN proteins (Supplementary Fig. 4a). Taken together, our results suggest that the interaction between VWCE and KICSTOR relies on the KPTN/ITFG2 heterodimer, aligning with the interaction map from the BioPlex database (Fig. 2d).

### VWCE is required for the lysosomal localization of GATOR1

VWCE knockdown impaired the interactions between KICSTOR, GATOR1, and GATOR2 (Fig. 3a–c and Supplementary Fig. 5a–g). Given that KICSTOR regulates mTORC1 by tethering GATOR1 to the lysosomes to exert its GAP activity towards Rags[8], we tested if VWCE is required for the lysosomal localization of KICSTOR and GATOR complexes. Interestingly, knockdown of VWCE decreased the level of GATOR1, but not GATOR2 or KICSTOR, on the immunopurified lysosomes (Fig. 3d). Accordingly, immunostaining experiments also revealed that the lysosomal localization of GATOR1, but not GATOR2 or KICSTOR, was impaired in VWCE knockdown cells (Fig. 3e–h and Supplementary Fig. 5h–k). These findings indicate that VWCE is required for KICSTOR to tether GATOR1 to lysosomes (Fig. 3i).

To further explore how VWCE influences the KICSTOR complex, we knocked down VWCE to examine the interactions among the KICSTOR components. The interactions of the KPTN/ITFG2 dimer, as well as SZT2-KPTN and SZT2-C12orf66, were not impacted by the VWCE deficiency (Fig. 3a and Supplementary Fig. 6a, b). Intriguingly,

knockdown of VWCE impaired the interaction between KPTN and C12orf66 (Supplementary Fig. 6c). In addition, a deficiency in VWCE reduced the proportion of KPTN proteins in fractions with higher molecular weights, as determined by SEC analysis (Supplementary Fig. 6d, e). These findings suggest that VWCE plays a role in preserving the appropriate conformation of the KICSTOR complex to tether GATOR1 to the lysosomes.

### VWCE inhibits cancer development via mTORC1 signaling

mTORC1 is a central hub for governing cellular metabolic homeostasis, and dysfunction of mTORC1 signaling has been implicated in cancers[2]. For instance, hyperactivation of mTORC1 has been found in up to 80% of human cancers[14]. Using the Cancer Genome Atlas (TCGA) expression database, we found that the expression level of VWCE was lower in tumors than in the adjacent normal tissues in multiple types of cancers, including breast cancer (BRCA), bile duct cancer (CHOL), head and neck cancer (HNSC), kidney chromophobe (KICH) and prostate cancer (PRAD) (Fig. 4a and Supplementary Table 2). We chose prostate cancer (PRAD) as a VWCE low-expression representative cancer type, and liver cancer (LIHC) as a VWCE high-expression control for subsequent functional studies. Consistent with the TCGA expression database, analysis based on the Cancer Cell Line Encyclopedia (CCLE) also indicates that the expression level of VWCE is much higher in the liver cancer cell lines HepG2 and Huh7 compared with the prostate cancer cell lines PC3, 22Rv1, and DU145 (Fig. 4b). This was also verified by RT-qPCR and immunoblotting assays (Fig. 4c, d).

To further evaluate the role of VWCE in cancer development, we first stably overexpressed VWCE in prostate and liver cancer cells, and measured the amino acid-regulated mTORC1 activity. Overexpression of VWCE suppressed amino acid-stimulated mTORC1 activation in prostate cancer cell lines, but not liver cancer cell lines (Fig. 4e, f and Supplementary Fig. 7a–f). Consistent with these results, the lysosomal localization of mTORC1 was impaired when VWCE was overexpressed in prostate cancer cells, but not in liver cancer cells (Fig. 4g, h and Supplementary Fig. 7g–n). More importantly, overexpression of VWCE inhibited the colony formation ability of prostate cancer cells (Fig. 4i and Supplementary Fig. 8a, b). In contrast, the proliferation of liver cancer cells was not affected by VWCE overexpression (Supplementary Fig. 8c, d). To further assess the role of VWCE in the development of prostate cancer, prostate cancer cells (PC3, 22Rv1, or DU145) overexpressing either a vector control or VWCE were subcutaneously transplanted into nude mice. Overexpression of VWCE inhibited the xenograft tumor growth of these prostate cancer cells (Fig. 4j–l and Supplementary Fig. 8e–j). Collectively, these data suggested that the lower expression level of VWCE is associated with prostate cancer development.

### Discussion

The ability to sense amino acid availability and adjust the metabolic programs accordingly is crucial for cell growth and survival. Amino acid signals are transduced from sensor proteins to the GATOR complexes. Despite the importance of the GATOR1 and GATOR2 complexes, the exact mechanisms of how amino acid signals regulate these two complexes are not well understood. The lysosomal localization of GATOR1 is critical for it to exert its GAP activity towards RagA/B. In this study, we identify VWCE as a KICSTOR-interacting protein that coordinates with KICSTOR to recruit GATOR1 to the lysosomes and modulates amino acid-dependent mTORC1 signaling. The identification of VWCE strengthens our understanding of the regulatory mechanism of GATOR1. Similar to the KICSTOR components, VWCE has no predicted lysosome-targeting sequence. Further analyses are required to determine how VWCE localizes to the lysosomes. Our findings suggest that VWCE may help maintain the appropriate conformation of the KICSTOR complex. However, detailed insights into the regulatory mechanism require further structural evidence. VWCE has been

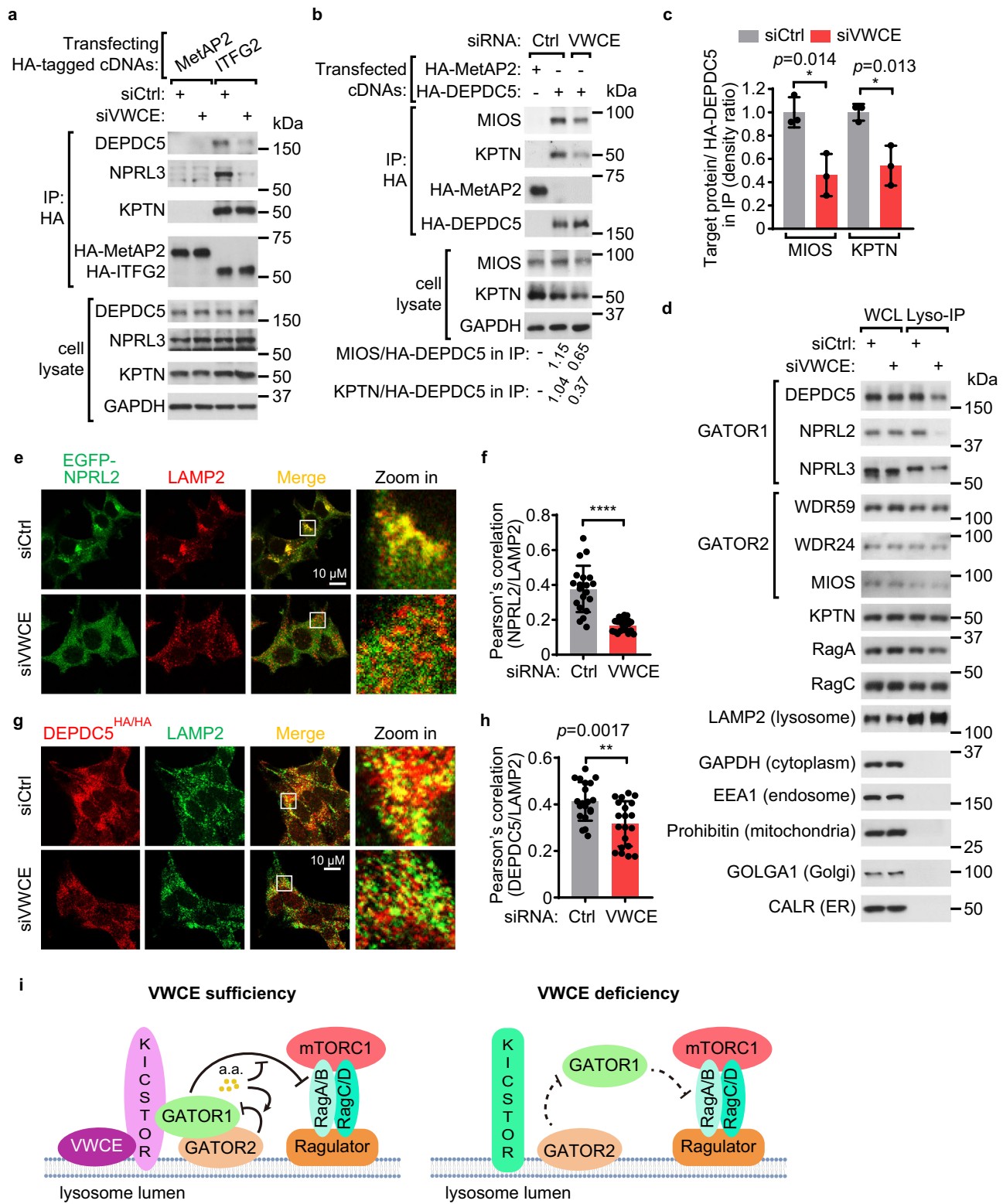

**Fig. 3 | Knockdown of VWCE suppresses the lysosomal localization of GATOR1.**
**a**, **b** Knockdown of VWCE impairs the GATOR1-KICSTOR (**a**) and GATOR1-GATOR2 (**b**) interactions. Cells were transfected with the indicated siRNAs and cDNAs. The immunoprecipitates and lysates were analyzed by immunoblotting. **c** The relative levels of immunoprecipitated MIOS/HA-DEPDC5 or KPTN/HA-DEPDC5 in (**b**) are quantified across three independent replicates. Data are mean ± s.d. *$p < 0.05$ (unpaired two-sided Student's *t*-test). **d** Knockdown of VWCE affects the lysosomal localization of GATOR1. Cells stably expressing 3×HA-tagged TMEM192 were transfected with siCtrl or siVWCE. Immunopurified lysosomes and cell lysates were

analyzed by immunoblotting. **e**, **g** Cells stably expressing EGFP-NPRL2 (**e**) or expressing endogenous 3×HA-tagged DEPDC5 (**g**) were transfected with the indicated siRNAs. Immunostaining experiments were performed. LAMP2 was used as a lysosomal marker. **f**, **h** Quantification of the co-localization between EGFP-NPRL2 and LAMP2 in (**d**) ($N = 20$) or 3×HA-DEPDC5 and LAMP2 in (**g**) ($N = 20$). Data are mean ± s.d. **$p < 0.01$; ****$p < 0.0001$ (unpaired two-sided Student's *t*-test). **i** A proposed model of how VWCE regulates amino acid-dependent mTORC1 signaling. The immunoblotting assays were independently replicated three times with consistent results. Source data are provided as Source data files.

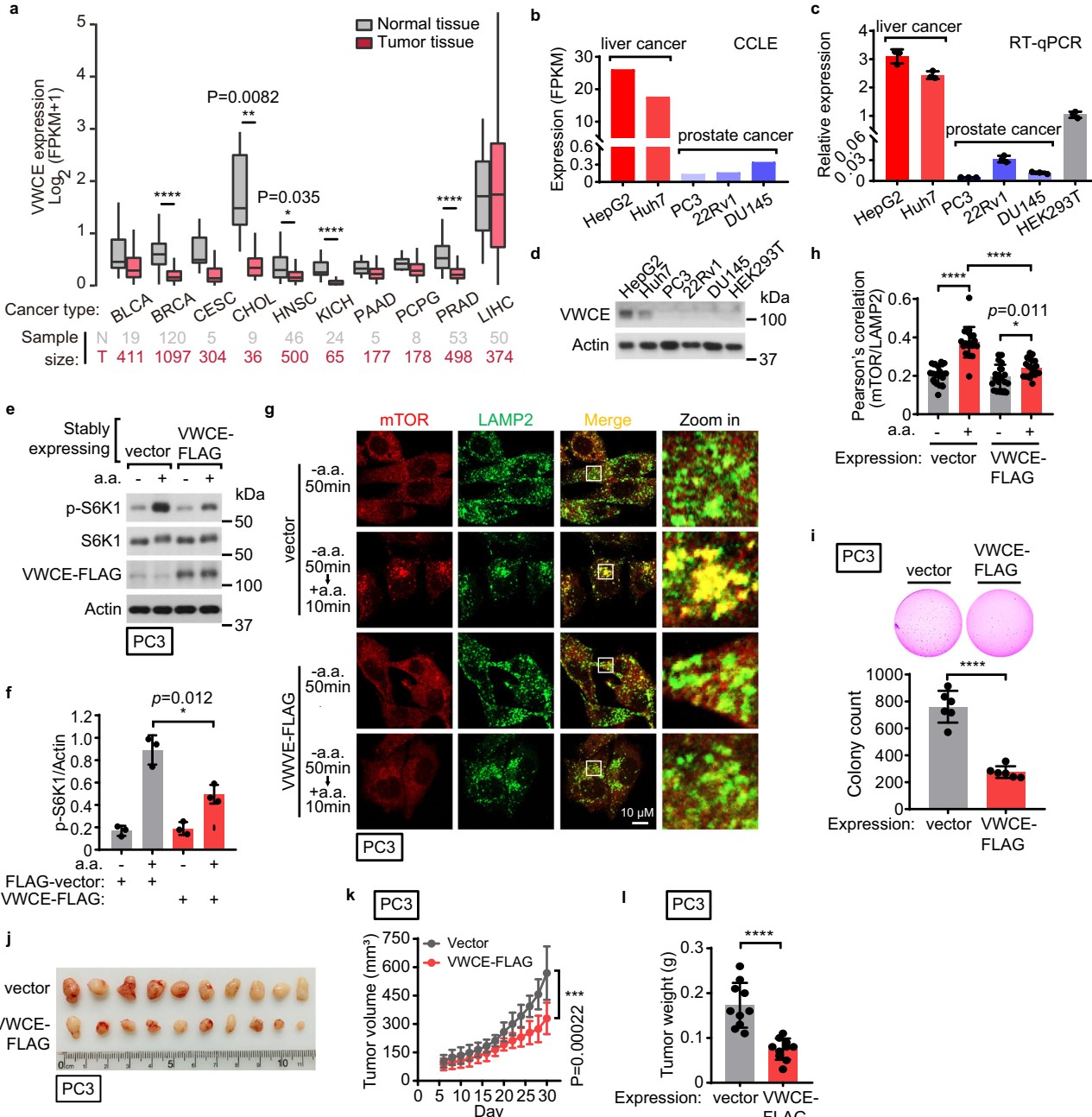

**Fig. 4 | VWCE inhibits cancer development through mTORC1 signaling.**
**a** Differential expression of VWCE in normal versus tumor tissues, as derived from the TCGA database. Sample sizes for normal (N) and tumor (T) tissues are indicated in the figure. *$p < 0.05$; **$p < 0.01$; ****$P < 0.0001$ (unpaired two-sided Student's *t*-test). The full names corresponding to the cancer types abbreviations are provided in Supplementary Table 2. **b** VWCE is highly expressed in liver cancer cell lines, but not in prostate cancer cell lines. The expression levels of VWCE in the indicated cancer cell lines were obtained from the CCLE database. **c, d** The expression levels of VWCE in the cancer cell lines were measured by RT-qPCR (**c**) and immunoblotting (**d**). Data are mean ± s.d. in (**c**) ($N = 3$). **e** Overexpression of VWCE inhibits amino acid-dependent mTORC1 activation in the PC3 cells. The cells stably expressing vector or FLAG-tagged VWCE were treated as indicated and lysed for immunoblotting. **f** The relative levels of p-S6K1/Actin in (**e**) are quantified across three independent replicates. Data are mean ± s.d. *$p < 0.05$ (unpaired two-sided Student's *t*-test). **g** Stable expression of VWCE inhibits the lysosomal localization of

mTOR in PC3 cells. The cells stably expressing vector or VWCE were treated as indicated and immunostained with anti-mTOR (red) and anti-LAMP2 (green) antibodies. **h** Quantification of co-localization between mTOR and LAMP2 in (**g**) ($N = 20$). Data are mean ± s.d. *$p < 0.05$; ****$p < 0.0001$ (unpaired two-sided Student's *t*-test). **i** Stable expression of VWCE inhibits anchorage-independent cell growth of PC3 cells. A total of 10,000 PC3 cells were seeded into each well of a 6-well cell culture plate. The number of colonies were quantified ($N = 6$). Data are mean ± s.d. ****$p < 0.0001$ (unpaired two-sided Student's *t*-test). **j**–**l** Overexpression of VWCE inhibits subcutaneous xenograft tumor growth of PC3 cells. The PC3 cells stably expressing vector or VWCE were subcutaneously injected into nude mice for xenograft growth. Tumor images (**j**), tumor volumes (**k**), and tumor weights (**l**) are shown. Tumor volumes and tumor weights are quantified ($N = 10$). Data are mean ± s.d., ***$p < 0.001$; ****$p < 0.0001$ (unpaired two-sided Student's *t*-test). Source data are provided as Source data files.

reported to function as a tumor suppressor in breast cancer via the induction of WDR1 expression[15]. It has also been reported to act as an oncogene in hepatocellular carcinoma by modulating the β-catenin pathway[12]. We reveal that VWCE acts as a tumor suppressor in prostate cancer by inhibiting the mTORC1 pathway. The expression level of VWCE is found to be lower in prostate cancer cells than in the adjacent normal tissues. Overexpression of VWCE in prostate cancer cells inhibits mTORC1 activation, anchorage-independent cell growth, and tumor development in nude mice. However, in liver cancers, the expression level of VWCE remains high in both tumors and normal tissues. It's plausible that the expression of other oncogenic factors may counteract the tumor-suppressing effect of VWCE in liver cancers.

## Methods

### Antibodies

Antibodies against pS6K1 T389 (9234, 1:1500 for immunoblotting), S6K1 (9202, 1:1000 for immunoblotting), pAKT1 S473 (4060, 1:1000 for immunoblotting), mTOR (2983, 1:1000 for immunoblotting, 1:400 for immunostaining), RagA (4357, 1:1000 for immunoblotting), RagC (3360, 1:1000 for immunoblotting), NPRL2 (37344, 1:1000 for immunoblotting), WDR59 (53385, 1:1000 for immunoblotting), MIOS (13557, 1:500 for immunoblotting), CALR (12238, 1:1000 for immunoblotting), HA (3724, 1:2000 for immunoblotting, 1:50 for immunostaining), EEA1 (3288, 1:1000 for immunoblotting) were from Cell Signaling Technology; antibodies against SEC13 (sc-514308, 1:1000 for immunoblotting), AKT1 (sc-5298, 1:1000 for immunoblotting), LAMP2 (sc-18822, 1:1000 for immunoblotting, 1:300 for immunostaining), GOLGA1 (sc-59820, 1:1000 for immunoblotting), Prohibitin (sc-28259, 1:1000 for immunoblotting) were from Santa Cruz Biotechnology; antibodies against KPTN (16094-1-AP, 1:1000 for immunoblotting), WDR24 (20778-1-AP, 1:1000 for immunoblotting), VDAC1 (55259-1-AP, 1:1000 for immunoblotting) were from ProteinTech; antibodies against VWCE (ab184772, 1:300 for immunoblotting), DEPDC5 (ab213181, 1:500 for immunoblotting), SEH1L (ab218531, 1:1000 for immunoblotting), GAPDH (ab128915, 1:5000 for immunoblotting) were from Abcam; antibody against Actin (ACTB; AC026, 1:20,000 for immunoblotting) was from Abclonal; antibodies against NPRL3 (HPA011741, 1:1000 for immunoblotting), FLAG (F7425, 1:3000 for immunoblotting, 1:50 for immunostaining; F1804, 1:1000 for immunoblotting), MYC (M4439, 1:2000 for immunoblotting) were from Sigma. The secondary antibody anti-rabbit HRP (7074, 1:20,000 for immunoblotting) was from Cell Signaling Technology and the secondary antibody anti-mouse HRP (A4416, 1:10,000 for immunoblotting) was from Sigma. Secondary antibodies anti-mouse Alexa Fluor 488 (green, A11029, 1:1000 for immunostaining) and 594 (red, A11032, 1:1000 for immunostaining), and anti-rabbit Alexa Fluor 594 (A11037, 1:1000 for immunostaining) and 405 (blue, A48254, 1:1000 for immunostaining) were from ThermoFisher.

### Cell lines and culture

HEK293T (#CRL-3216) and HEK293E (#CRL-10852) were from ATCC, HepG2 was from Prof. Lei Chen (Peking University), Huh7 was from Prof. Xiao-Wei Chen (Peking University), and prostate cancer cell lines (PC3, 22Rv1, and DU145) were from Prof. Hong Wu (Peking University). HEK293T, HEK293E, DU145, HepG2, and Huh7 were kept in high glucose Dulbecco's modified Eagle medium (DMEM, Gibco C11995500BT). PC3 was kept in Ham's F-12K (Gibco 21127022). 22Rv1 was kept in RPMI 1640 (Gibco 11875093). Cells were maintained in the cell culture medium supplemented with 10% fetal bovine serum (FBS), 100 U/mL penicillin, and 100 mg/mL streptomycin in a humidified incubator at 37 °C and 5% $CO_2$.

### Cell treatments

For amino acid starvation, when the cell confluence was ~80%, cells were washed with phosphate-buffered saline (PBS) and incubated in amino acid-free DMEM (US Biologicals, D9800-27) supplemented with 4.5 g/L glucose and 10% dialyzed FBS (dFBS; Biological Industries, 04-011-1) for 50 min unless otherwise stated. For amino acid restimulation, cells were starved of amino acids for 50 min and re-supplemented with amino acids for 10 min unless otherwise stated. The 50× glutamine-free amino acids mixture (Gibco 11130-051) and 100× glutamine solution (Gibco 25030-081) were directly diluted into the amino acid-free medium for restimulation. Handmade DMEM without a specific amino acid (leucine, isoleucine, valine, lysine, or arginine) was prepared based on the standard recipe and supplemented with 20 mM HEPES (Thermo Fisher Scientific, 15630080) and 10% dFBS. In addition, 100× handmade leucine (80 mM), isoleucine (80 mM), valine (80 mM), lysine (80 mM), or arginine (40 mM) solution was directly diluted into the corresponding amino acid-free medium for restimulation. For serum starvation, cells were washed with PBS and incubated with standard DMEM without FBS for 2 h. For serum restimulation, cells were starved of serum for 2 h and supplemented with 10% FBS for 20 min.

### Screening for the regulators of mTORC1 by RNAi

Based on the BioPlex and BioGRID databases, proteins interacting with at least two components of the amino acid-regulated mTORC1 pathway were selected as candidates. For each candidate, two small interfering RNAs (siRNAs) were synthesized and transfected into HEK293T cells. Seventy-two hours after transfection, cells were treated by amino acid starvation or restimulation, and lysed for immunoblotting analysis.

### cDNA and siRNA transfection

For overexpression, cDNAs were cloned into pCDNA3.3, pRK5, or pBOBI vector. Plasmids were transfected into cells using polyethylenimine (PEI) in serum-free DMEM when the cell confluence was about 70%, and the medium was replaced 6 h after transfection. Cells were treated and harvested 48 h post-transfection.

For siRNA-mediated knockdown, siRNAs were transfected into cells with RNAiMAX (Thermo Fisher, 13778150) reagent in opti-MEM (Gibco 31985088) when the cell confluence was about 70%. Cells were treated and harvested 72 h after transfection. If cells were transfected with both siRNAs and cDNAs, siRNAs were firstly delivered into cells, followed by transfection of cDNAs 6 h later. Cells were harvested 72 h after siRNA transfection. The target sequences for siRNAs were: siCtrl (targeting luciferase): CGUACGCGGAAUACUUCGATT; siVWCE: GCU-GUGACCUUACCUGCAATT; siWDR59: GCGCGAGGAGCAGCGAAATT; siSZT2: GGAUCGUUGGAAACUAAGATT.

### Establishment of stable cell lines

The plasmids for stable expression of cDNAs or short hairpin RNAs (shRNAs) were generated using the pLJM1 or pLKO.1 vector. HEK293T cells were transfected with pLJM1 or pLKO.1-based plasmids, together with plasmids for virus packaging (pMDLg/pRRE, pRSV-Rev, and pCI-VSVG) when the cell confluence was about 70%. Cell culture medium was replaced 6 h after transfection. The medium was harvested and filtered with a 0.45 μM filter after another 48 h to collect lentivirus. For virus infection, cells were cultured with 500 μL virus mixed with 500 μL medium and 8 μg/ml polybrene (Sigma, H9268) in a 12-well plate for 24 h. The infected cells were reseeded in medium supplemented with 2 μg/mL puromycin (Gibco, A1113803) or 10 μg/mL blasticidin (Invivo Gene, ant-bl-05) until all the uninfected control cells died (at least 48 h)[16]. The target sequence for shRNA was shVWCE: AGGCTGCTCTCTTGACGACAA.

### Generation of knockout cells

SZT2, MIOS, and NPRL3 knockout HEK293T cells were generated previously by our lab through the CRISPR/Cas9 approach. For the generation of other knockout cell lines, single-guide RNAs (sgRNAs) targeting

specific gene genomic sequences were synthesized and cloned into the lentiCRISPR V2 vector. HEK293T cells were transfected with these constructs and subjected to selection with 2 µg/mL puromycin 24 h post-transfection. After 72 h of selection, the cells were individually cloned and expanded for genomic identification via DNA sequencing. The sgRNA oligos used for KPTN, ITFG2, and C12orf66 were as follows: sgKPTN-sense: caccgGCGCAACGGACAAGGCCCCG; sgKPTN-antisense: aaacCGGGGCCTTGTCCGTTGCGCc; sgITFG2-sense: caccgGGTGGGAGA CACCAGCGGGA; sgITFG2-antisense: aaacTCCCGCTGGTGTCTCCCACCc sgC12orf66-sense: caccgCGAGAGGCCAACAAGAGCGC; sgC12orf66-anti-sense: aaacGCGCTCTTGTTGGCCTCTCGc.

### Generation of the knock-in cells

HEK293T cells expressing endogenous 3×HA-DEPDC5 were generated previously by our lab[17]. HEK293T cells expressing endogenous 3×FLAG-DEPDC5 and VWCE-FLAG-P2A-EGFP were generated using a homologous recombination repair strategy based on the CRISPR/Cas9 approach. For the generation of 3xFLAG-DEPDC5 knock-in cells, HEK293T cells were transfected with the PX459 construct expressing sgRNA that targets DEPDC5. This was combined with a double-stranded DNA template containing the 3xFLAG-coding sequence, a linker, and silent mutations near the start codon of the DEPDC5 gene. For the generation of VWCE-FLAG-P2A-EGFP knock-in cells, the sgRNA targeting the region near the stop codon of VWCE (sgVWCE KI) was cloned into a lentiCRISPR V2 vector. HEK293T cells were co-transfected with this constructed vector and a linear double-stranded DNA donor. To prepare the donor, homologous fragments of approximately 500 bp flanking the sgRNA target site and the insertion fragment containing FLAG-loxP-P2A-EGFP-loxP were amplified by PCR and ligated using a homologous recombination kit (Vazyme, C112-01). Following transfection, cells were selected with puromycin and individually cloned. The clones were identified by PCR and DNA sequencing.

To generate HEK293T cells with endogenously C-terminally FLAG-tagged VWCE (VWCE^FLAG/FLAG), HEK293T cells expressing endogenous VWCE-FLAG-loxP-P2A-EGFP-loxP were transfected with a Cre construct, which was backboned into a pLJM1 vector. After transfection, the cells were selected with 10 µg/mL blasticidin for 72 h. After the selection period, the cells were individually cloned. A homozygous VWCE-FLAG knock-in clone was identified through PCR and DNA sequencing. The sgRNA oligos used for the knock-in cells were as follows sgVWCE KI-sense: caccgACAGTGACCTCCTTACATGG; sgVWCE KI-antisense: aaacCCATGTAAGGAGGTCACTGTc; sgDEPDC5 KI-sense: caccgTGCAAGATGAGAACAACAA; sgDEPDC5 KI-antisense: aaacTT GTTGTTCTCATCTTGCAc. The sequence of the donor for the 3×FLAG-DEPDC5 knock-in cell was: CAAGCTTGGAACAGCTAAAGGGAAAAACA GTGCAAGATGGACTACAAAGACCATGACGGTGATTATAAAGATCATG ACATTGATTACAAGGATGACGATGACAAGGCGGCCGCAGGCCGTACG ACGAAAGTCTACAAACTCGTCATCCACAAGAAGGGCTTTGGG.

### Flow cytometry analysis

HEK293T cells, which express endogenous VWCE-FLAG-P2A-EGFP, were transfected with either siCtrl or siVWCE. Seventy-two hours post-transfection, the cells were trypsinized and resuspended in PBS containing 2% FBS. The cell suspensions were then analyzed using a Beckman Astrios EQ flow cytometer. Refer to Supplementary Fig. 2 for the gating strategy applied in this analysis.

### RNA isolation and RT-qPCR

Cells seeded in a 12-well plate were lysed with TRIzol reagent (TransGen ET111-01). According to a standard procedure, total RNA was extracted by chloroform and precipitated by isopropanol. 1000 ng RNA was subjected to reverse transcription using a commercial kit (TransGen, AT311). Real-time quantitative PCR (RT-qPCR) was carried out using SYBR green. Quantification of the transcript

levels was normalized to Actin. RT-qPCR primers used are VWCE forward: GTGTAGATGTAAACGAGTGTCGG; VWCE reverse: GTCG GCATGTGCATAGGAAG; SZT2 forward: CTGGTCAGTATGATTCGTCA GGG; SZT2 reverse: AATTCCACATTGGGCACATGG; Actin forward: CATGTACGTTGCTATCCAGGC; Actin reverse: CTCCTTAATGTCAC GCACGAT.

### Cell lysis, immunoprecipitation, and immunoblotting

To analyze total protein levels or phosphorylated protein levels in whole cell lysates, cells were lysed with 100 µL Triton X-100 lysis buffer (1% Triton X-100, 40 mM HEPES pH 7.4, 10 mM pyrophosphate, 10 mM β-glycerol phosphate, 2.5 mM $MgCl_2$, supplemented with EDTA-free protease inhibitor cocktail (Roche 4693132001) and phosphatase inhibitors (Bimake B15002)) when the cell confluence was about 80%. The lysates were clarified by centrifugation at $20,000 \times g$ for 10 min at 4 °C. The supernatants were mixed with 4× Laemmli sample buffer (Bio-Rad) supplemented with β-mercaptoethanol and boiled for 10 min at 100 °C.

For immunoprecipitation experiments, cells in a 10-cm dish were washed once with ice-cold PBS and lysed with 1 mL Triton X-100 lysis buffer when the cell confluence was about 80%. Cell lysates were centrifuged, and the supernatants were incubated with pre-washed FLAG or HA agarose beads (Sigma) for 2 h at 4 °C with gentle rotation. The beads were washed three times with Triton X-100 lysis buffer containing 150 mM NaCl and boiled with 2× Laemmli sample buffer (Bio-Rad) supplemented with β-mercaptoethanol at 100 °C for 10 min.

For immunoblotting experiments, protein samples were separated by 10% SDS-PAGE and transferred onto PVDF membranes at 100 V for 90 min. For SZT2, the transfer time was extended to 6 h. The membranes were blocked in 5% non-fat milk dissolved in TBST for 1 h at room temperature. The blocked membranes were then incubated with primary antibody overnight at 4 °C, followed by incubation with secondary antibody for 1 h at room temperature. The membranes were developed with the enhanced chemiluminescence method, and visualized by Kodak films. The unprocessed scans of the blot data were provided in Source data.

### Size-exclusion chromatography (SEC)

A near confluent 15-cm plate of cells was washed with ice-cold PBS and lysed with 500 µL Triton X-100 lysis buffer. The lysates were then centrifuged at $20,000 \times g$ for 30 min at 4 °C. 500 µL of the cleared supernatants were loaded on a Superose 6 Increase 10/300 GL Column (GE Healthcare, 29-0915-96). The elution was then fractioned (0.5 mL per fraction) in column buffer (40 mM HEPES pH 7.4, 150 mM NaCl) using an ATKA purifier (AKTA pure, GE Healthcare). The fractions were mixed with 4× Laemmli sample buffer supplemented with β-mercaptoethanol and boiled for 10 min at 100 °C.

### Immunostaining and confocal microscopy

~70,000 cells were seeded in each well of a 24-well plate containing 14-mm poly-D-lysine-coated coverslips (Sigma, P1149) and cultured overnight. Cells treated under each condition were washed once with ice-cold PBS and fixed in 4% paraformaldehyde solution for 20 min at room temperature. Cells were then washed three times with PBS and permeabilized for 10 min with 0.1% Triton X-100 dissolved in PBS. The coverslips were blocked for 1 h with 5% BSA solution (dissolved in PBS), and incubated overnight at 4 °C with primary antibodies diluted in 5% BSA solution (LAMP2, 1:300; mTOR, 1:400; Flag, 1:100). The coverslips were then washed three times with PBS and incubated with fluorophore-labeled secondary antibodies diluted 1:1000 in 5% BSA solution. Incubation was carried out for 1 h in the dark at room temperature. The coverslips were washed five times with PBS, taken out from the 24-well plate, and fixed onto slides with ProLong diamond antifade mountant containing DAPI (ThermoFisher Scientific, P36966). The slides were imaged with laser scanning confocal microscopes

(Zeiss LSM 710 NLO & DuoScan System, LSM 880 META UV/Vis or LSM980 with Airyscan2). Quantifications were performed using Fiji software (version 1.0) coupled with the Colocalization_Finder plugin.

## Lysosome purification and the trypsin treatment

Lysosomes were isolated with a rapid immunopurification method (LysoIP) following a previous study with minor modifications[13]. All cells utilized for lysosome purification stably expressed TMEM192-3×HA. The cells at 70% confluence in a 15-cm dish were prepared for each LysoIP. All equipment and buffers were ice-cold, and the assay was performed on ice. Cells were washed once with PBS, scraped off in PBS, and centrifuged at $1000 \times g$ for 2 min at 4 °C. Cell pellets were resuspended with 950 μL PBS (supplemented with protease and phosphatase inhibitors). 12.5 μL resuspended cells were lysed with Triton X-100 lysis buffer for whole cell lysates. The remaining cells were gently homogenized with 20 strokes using a Dounce homogenizer. The homogenates were centrifuged at $1000 \times g$ for 2 min at 4 °C and the supernatants were incubated with 150 μL pre-washed anti-HA magnetic beads (Thermo Fisher Scientific) for 3.5 min with rotation at 4 °C. The immunoprecipitates were washed three times with PBS, lysed in Triton X-100 lysis buffer on ice for 10 min, and vortexed three times. The lysosomal lysates were mixed with 4 × Laemmli sample buffer and boiled for 10 min at 100 °C for immunoblotting. For the trypsin treatment, immunoprecipitates washed with PBS were incubated with 0, 2.5, or 10 μg of trypsin. The mixture was then incubated at 37 °C for 5 min before being processed for immunoblotting.

## Soft agar colony formation

1.5 mL warmed-up bottom-layer agarose [1:1 mixture of 1.2% low melting point agarose (Sigma A9045) and 2 × DMEM supplemented with 20% FBS] was added to each well of a 6-well plate and cooled down at room temperature for solidification. 5000 to 10,000 cells were suspended in 1.5 mL warmed-up top-layer agarose (1:1 mixture of 0.7% low melting point agarose and culture medium) for each well. 0.7% low melting point agarose was 1:1 mixed with the cell culture medium (RPMI 1640 for 22Rv1, F-12K for PC3, DMEM for other cell types) supplemented with 40% FBS. After the top layer was added, the plates were immediately cooled down at 4 °C for 10 min, and transferred to an incubator. 100 μL medium was added to each well every four days to keep the agarose wet. About 2–4 weeks later, cell colonies in each well were stained with 1 mL crystal violet (0.005% in 5% methanol) for 60 min at 37 °C, washed once with PBS, imaged and quantified by Fiji software (version 1.0).

## Subcutaneous xenograft tumor growth

The animal study protocols were evaluated and approved by the Institutional Animal Care and Use Committee (IACUC, FT-LiuY-5) at Peking University. Six-week-old male BALB/c nude mouse were obtained from Charles River and housed in a specific pathogen-free facility. The mice were kept under ~20 °C with ~50% humidity in a 14-h light/10-h dark cycle. Prostate cancer cells were trypsinized, washed twice with PBS, and resuspended in PBS. For each injection, 100 μL of the cell suspension was mixed with 100 μL of Matrigel (Corning) and subcutaneously injected into the right flank of each mouse using an insulin syringe (BD Biosciences). The number of cells in each injection varied with different cell lines: 3 million for PC3, 2 million for 22Rv1, and 2.5 million for DU145. Xenografts were measured with a digital caliper every other day since the 5th day after transplantation (tumor volume = width$^2$ × length × 0.523). On the final day, the xenografts were dissected, snapped, and weighted. The tumor size did not exceed 15 mm in its largest diameter. According to the guidelines of the IACUC at Peking University, all tumors were maintained at a size not exceeding 20 mm in their largest dimension. All xenograft tumors adhered to this size limitation.

## Statistical analysis

All graphs with error bars or statistical significance in this study were generated by Graphpad Prism. Two-tailed Student's *t*-test was performed in this study to compare values between different groups. Results are presented as mean ± s.d. of at least three replicates unless otherwise stated. Biological replicates (N) are indicated in the figure legends. Statistical significances are as follows: n.s., not significant; *$P < 0.05$, **$P < 0.01$, ***$P < 0.001$, ****$P < 0.0001$. In a box and whisker plot, the box's upper and lower ends represent the upper and lower quartiles, spanning the interquartile range. The median is indicated by a horizontal line within the box. Whiskers extend from the box to the highest and lowest observations. The mRNA expression data from The Cancer Genome Atlas (TCGA) were analyzed using R (version 4.0.3) and R packages.

## Reporting summary

Further information on research design is available in the Nature Portfolio Reporting Summary linked to this article.

## Data availability

Details about VWCE's molecular features were obtained from the UniProt database (https://www.uniprot.org/). The interaction information for siRNA screening in Supplementary Table 1 was from BioGRID (https://thebiogrid.org/) and BioPlex (https://bioplex.hms.harvard.edu/) databases. FPKM-normalized mRNA expression data for Fig. 4a, derived from The Cancer Genome Atlas (TCGA), was acquired from the UCSC Xena data hub (https://xenabrowser.net/hub/). The mRNA expression data of cancer cell lines for Fig. 4b, from the CCEL project, was obtained from the DepMap Portal (https://depmap.org/portal/download/all/). All other data supporting the findings of this study are available from the corresponding authors upon request. Source data are provided with this paper.

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

## Acknowledgements

Y.L. was supported by grants from the National Key R&D Program of China (No. 2022YFA0806502), the National Natural Science Foundation of China (grants nos. 31925012, 92157301, and 9225430010), and the HHMI International Research Scholar Program (grant no. 55008739). This work was also supported by the Peking-Tsinghua Center for Life Sciences. Y.L. acknowledges support from the Tencent Foundation through the XPLORER PRIZE and the New Cornerstone Investigator Program.

## Author contributions

T.Z. and Y.L. conceived the study, designed the experiments, analyzed the data, and wrote the manuscript. T.Z. performed the experiments with assistance from Y.G., C.X., D.W., and J.G. J.G. performed bioinformatic analysis.

## Competing interests

The authors declare no competing interests.
