## [Peer Review File · Nature Communications]

VWCE modulates amino acid-dependent mTOR signaling and coordinates with KICSTOR to recruit GATOR1 to the lysosomesReviewers' comments:

Reviewer #1 (Remarks to the Author):

By knockdown of VWCE, they showed that S6K remains phosphorylated despite absence of amino acids, indicating VWCE is a negative regulator of mTORC1. Next, they performed epistasis analysis and found that VWCE associates with KICSTOR and is downstream of GATOR2 but upstream of GATOR1 and Rag. By lysosome immune-capture, they found that VWCE localizes to this compartment in an aa-insensitive manner but is independent of KICSTOR. They also found that VWCE is required for tethering GATOR1 to KICSTOR at the lysosomes. They next analyzed different cancers and found that VWCE expression is low in prostate cancers and that overexpression prevented colony formation and diminished mTORC1 signaling. Based on these findings, they concluded that “VWCE modulates aa-dependent mTOR signaling and coordinates with KICSTOR to recruit GATOR1 to lysosomes.”

The studies are interesting and identify another protein that is involved in mTORC1 regulation by amino acids via lysosomal localization. The data are mostly robust. However, most of the analyses are done on overexpressed VWCE. The knockdown of VWCE (protein expression) was also not verified in all of the experiments. Thus it is not clear how relevant is VWCE interaction with KICSTOR and its regulation of mTORC1 signaling at endogenous levels. The mechanisms as to how VWCE mediate KICSTOR recruitment of GATOR1 were also only superficially addressed. Lastly, the role of VWCE as a possible tumor suppressor is interesting but was not fully interrogated either. Hence, the studies are highly preliminary.

Reviewer #2 (Remarks to the Author):

The authors present a model in which von Willebrand factor C and EGF domain-containing protein (VWCE) acts as a negative regulator of mTORC1 activation by amino acids through coordination of KICSTOR function. VWCE was identified as a potential regulator of mTORC1 regulation by amino acids by mining data from the published BioPlex interaction proteomics dataset. Using RNA interference and a variety of biochemical and cell biology approaches the authors suggest that VWCE coordinates KICSTOR function, through an unknown mechanism, to recruit GATOR1 to the lysosome surface. By mining The Cancer Genome Atlas they identify cancers with decreased VWCE expression, which include prostate cancers, and use the Cancer Cell Line Encyclopedia to select prostate cancer cell lines for functional analysis of VWCE. Consistent with their model, increasing expression of VWCE in prostate cancer lines restored mTORC1 amino acid sensitivity and inhibited anchorage-independent cell growth.

While indeed there is value in elucidating mechanisms by which KICSTOR coordinates GATOR1 complex function, in my opinion the model presented here is too preliminary to be published in its current form. My concerns are outlined below.

1) I am concerned by the lack of attention given to the molecular features of the VWCE protein itself, and whether it is plausible that it could perform the functions suggested in the model. For example, VWCE contains a predicted N-terminal signal sequence, which likely directs it to the ER lumen upon translation. Furthermore, it has predicted EGF-like domains, which contain several cysteines capable of disulfide bonds, features that almost exclusively exist in luminal or extracellular proteins / regions of proteins. Therefore, the likelihood that VWCE exists cytoplasmically, which the model as presented would require, seems very low. As it stands, the authors did not present any evidence that VWCE exists on the cytoplasmic face of lysosomes. If the authors believe that VWCE is in fact a cytoplasmic-facing protein, then they must provide a compelling argument in favour of this and ideally provide experimental evidence to support it — e.g., protease treatment of immuno-isolated lysosomes. If VWCE is targeted to the lysosome lumen and nonetheless performs the functions suggested in the paper, this needs to be explicitly discussed, and some sort of 'inside-out' mechanism would need to be invoked.

2) Why was immunofluorescence of the VWCE-FLAG construct expressed in cells never shown? Was this experiment attempted?

3) What were the criteria that were used to identify the 'candidate interacting proteins' from the BioPlex dataset listed in Ext Fig 1? It is stated that these are proteins that can 'interact with at least two known components of the mTORC1 pathway' but it is not defined what constitutes an 'interaction' and what components of the mTORC1 were considered. Please provide details.

4) Some more experimental detail could be provided with respect to starvation / restimulation experiments. For example, the amino acid-free media that was used for starvation does not contain glucose — was glucose added or were cells starved of glucose as well as amino acids? When serum starvation was tested, was this using dialyzed FBS, and was this in standard or amino acid-free DMEM?

5) In many instances the differences observed by Western blot are subtle, yet the conclusions drawn by the authors are absolute. The conclusions would be more convincing if data from independent ($> / = 3$) Western blot experiments were quantified and summarized.

6) The immunofluorescence images are small and difficult to interpret. If possible, please make these images larger, especially the magnified insets.

Reviewer #3 (Remarks to the Author):

The manuscript by Zhao et al describes VWCE, a novel regulator of the mTORC1 signaling activity. Mostly via protein interaction and cell imaging experiments, the authors provided data supporting a model where VWCE interacts with the KICSTOR complex on the lysosome to recruit GATOR1 and suppress mTORC1 activity. The authors also presented functional data demonstrating a tumor-suppressing role of VWCE in prostate cancer cells.

Suggestions for revision:

In Fig. 3b and Extended Fig. 3a-c, bands still show significant interactions between GATOR2 component and GATOR1 or KICSTOR component (e.g. MISO and DEPDC5, MISO and SZT2) under the VWCE knockdown condition. Therefore, I am not sure if the authors can conclude that interactions with GATOR2 complex is disrupted by VWCE knockdown. If anything, the effect seems to be subtle.

In Fig. 3c, the decrease of NPRL2 under siVWCE is convincing, but the decrease of NPRL3 is subtle. How about the other component DEPDC5? It is not shown in the figure. Also, in Fig. 3d-e, only NPRL2 decrease is shown. Is siVWCE affecting specifically NPRL2 or the whole GATOR1 complex?

The authors state that “VWCE may serve as a potential therapeutic target for the treatment of prostate cancer”. Besides the in vitro colony formation assay, xenografting of prostate cancer cells overexpressing VWCE onto nude mice may better support this point.

The authors should discuss why VWCE exert different effects on prostate cancer and liver cancer, and how it may shed light onto the molecular functions of VWCE.

Other minor points:

Fig. 2g. Why is LAMP2 expression so low in the whole cell lysates?

Fig. 2h. Immunoblotting of SZT2 needs to be shown to confirm its efficient knockdown.

Fig. 4g. Suppression of mTORC1 activity by VWCE overexpression in the DU145 cell line is not obvious. Similarly, in Fig. 4i, mTORC1 lysosomal localization was not altered significantly in DU145 cells.

Reviewers' comments:

We thank all the reviewers for the encouraging remarks on our work. We are particularly grateful for the constructive comments that have helped us to improve the study.

Reviewer #1:

By knockdown of VWCE, they showed that S6K remains phosphorylated despite absence of amino acids, indicating VWCE is a negative regulator of mTORC1. Next, they performed epistasis analysis and found that VWCE associates with KICSTOR and is downstream of GATOR2 but upstream of GATOR1 and Rag. By lysosome immune-capture, they found that VWCE localizes to this compartment in an aa-insensitive manner but is independent of KICSTOR. They also found that VWCE is required for tethering GATOR1 to KICSTOR at the lysosomes. They next analyzed different cancers and found that VWCE expression is low in prostate cancers and that overexpression prevented colony formation and diminished mTORC1 signaling. Based on these findings, they concluded that “VWCE modulates aa-dependent mTOR signaling and coordinates with KICSTOR to recruit GATOR1 to lysosomes.”

The studies are interesting and identify another protein that is involved in mTORC1 regulation by amino acids via lysosomal localization. The data are mostly robust. However, most of the analyses are done on overexpressed VWCE. The knockdown of VWCE (protein expression) was also not verified in all of the experiments. Thus it is not clear how relevant is VWCE interaction with KICSTOR and its regulation of mTORC1 signaling at endogenous levels. The mechanisms as to how VWCE mediate KICSTOR recruitment of GATOR1 were also only superficially addressed. Lastly, the role of VWCE as a possible tumor suppressor is interesting but was not fully interrogated either. Hence, the studies are highly preliminary.

We appreciate the constructive feedback provided by the reviewer. We have followed the reviewer's advice and conducted the following sets of experiments:

1. The knockdown of VWCE at the endogenous level (protein expression) was verified using FACS and immunoblotting assays (Extended Data Figs. 1d-f).
2. We generated VWCE-FLAG knock-in cells, which allowed us to immunoprecipitate endogenous VWCE. In the revised manuscript, we demonstrated that VWCE interacts with the KICSTOR component at the endogenous level (Extended Data Figs. 2a, b).
3. We further explored the mechanisms through which VWCE facilitates the recruitment of GATOR1 by KICSTOR. Our findings suggest that VWCE relies on the KPTN/ITFG2 heterodimer to interact with KICSTOR (Extended Data Fig. 3). VWCE may play a role in maintaining the correct conformation of the KICSTOR complex to anchor GATOR1 onto lysosomes (Extended Data Fig. 5).
4. We conducted xenograft assays in nude mice, which revealed that VWCE acts as a tumor suppressor in prostate cancer (Figs. 4j-l and Extended Data Figs. 7e-j).

Reviewer #2:

The authors present a model in which von Willebrand factor C and EGF domain-containing protein (VWCE) acts as a negative regulator of mTORC1 activation by amino acids through coordination of KICSTOR function. VWCE was identified as a potential regulator of mTORC1 regulation by amino acids by mining data from the published BioPlex interaction proteomics dataset. Using RNA interference and a variety of biochemical and cell biology approaches the authors suggest that VWCE coordinates KICSTOR function, through an unknown mechanism, to recruit GATOR1 to the lysosome surface. By mining The Cancer Genome Atlas they identify cancers with decreased VWCE expression, which include prostate cancers, and use the Cancer Cell Line Encyclopedia to select prostate cancer cell lines for functional analysis of VWCE. Consistent with their model, increasing expression of VWCE in prostate cancer lines restored mTORC1 amino acid sensitivity and inhibited anchorage-independent cell growth.

While indeed there is value in elucidating mechanisms by which KICSTOR coordinates GATOR1 complex function, in my opinion the model presented here is too preliminary to be published in its current form. My concerns are outlined below.

1) I am concerned by the lack of attention given to the molecular features of the VWCE protein itself, and whether it is plausible that it could perform the functions suggested in the model. For example, VWCE contains a predicted N-terminal signal sequence, which likely directs it to the ER lumen upon translation. Furthermore, it has predicted EGF-like domains, which contain several cysteines capable of disulfide bonds, features that almost exclusively exist in luminal or extracellular proteins / regions of proteins. Therefore, the likelihood that VWCE exists cytoplasmically, which the model as presented would require, seems very low. As it stands, the authors did not present any evidence that VWCE exists on the cytoplasmic face of lysosomes. If the authors believe that VWCE is in fact a cytoplasmic-facing protein, then they must provide a compelling argument in favour of this and ideally provide experimental evidence to support it — e.g., protease treatment of immuno-isolated lysosomes. If VWCE is targeted to the lysosome lumen and nonetheless performs the functions suggested in the paper, this needs to be explicitly discussed, and some sort of ‘inside-out’ mechanism would need to be invoked.

We thank the reviewer for the constructive comments. We have followed the reviewer’s advice to treat immuno-isolated lysosomes with trypsin. Our results indicated that lysosomal VWCE can be digested by trypsin, demonstrating that VWCE is localized on the cytosolic surface of lysosomes (Extended Data Fig. 2j).

2) Why was immunofluorescence of the VWCE-FLAG construct expressed in cells never shown? Was this experiment attempted?

Following the reviewer's advice, we conducted an immunostaining assay. Our findings revealed that VWCE is localized both in the nucleus and the cytosol. Within the cytosol, VWCE is constitutively present on lysosomes, regardless of the availability of amino acids (Extended

Data Figs. 2h, i).

3) What were the criteria that were used to identify the ‘candidate interacting proteins’ from the BioPlex dataset listed in Ext Fig 1? It is stated that these are proteins that can ‘interact with at least two known components of the mTORC1 pathway’ but it is not defined what constitutes an ‘interaction’ and what components of the mTORC1 were considered. Please provide details.

Based on the reviewer's advice, we've included detailed information in the revised manuscript. Details of the bait components of mTORC1 and the candidate prey proteins are provided in Supplementary Information Table 1.

4) Some more experimental detail could be provided with respect to starvation / restimulation experiments. For example, the amino acid-free media that was used for starvation does not contain glucose — was glucose added or were cells starved of glucose as well as amino acids? When serum starvation was tested, was this using dialyzed FBS, and was this in standard or amino acid-free DMEM?

We appreciate the reviewer for highlighting these points and guiding us to enhance the clarity of our study. The media used for amino acid starvation was supplemented with 4.5 g/L glucose. Meanwhile, the media used for serum starvation was standard DMEM without FBS. Detailed descriptions have been added to the 'Cell Treatments' section of the Methods."

5) In many instances the differences observed by Western blot are subtle, yet the conclusions drawn by the authors are absolute. The conclusions would be more convincing if data from independent ($> / = 3$) Western blot experiments were quantified and summarized.

In line with the reviewer's suggestions, we have quantified the Western blots, which have not yielded significantly noticeable effects. The results have been summarized based on three independent experiments (Figs. 3b, c; 4f and Extended Data Figs. 2b; 4b-g; 6b, d).

6) The immunofluorescence images are small and difficult to interpret. If possible, please make these images larger, especially the magnified insets.

We have followed the reviewer's advice to enlarge the magnified insets.

Reviewer #3:

The manuscript by Zhao et al describes VWCE, a novel regulator of the mTORC1 signaling activity. Mostly via protein interaction and cell imaging experiments, the authors provided data supporting a model where VWCE interacts with the KICSTOR complex on the lysosome to recruit GATOR1 and suppress mTORC1 activity. The authors also presented functional data demonstrating a tumor-suppressing role of VWCE in prostate cancer cells.

Suggestions for revision:

In Fig. 3b and Extended Fig. 3a-c, bands still show significant interactions between GATOR2 component and GATOR1 or KICSTOR component (e.g. MISO and DEPDC5, MISO and SZT2) under the VWCE knockdown condition. Therefore, I am not sure if the authors can conclude that interactions with GATOR2 complex is disrupted by VWCE knockdown. If anything, the effect seems to be subtle.

We appreciate the reviewer for posing this question. To strengthen our conclusion, we have quantified these interaction data and compiled the results from three independent experiments (Fig. 3b, c, and Extended Data Figs. 4b-g). In the revised manuscript, we have also amended the language by substituting “disrupt” with “impair”.

In Fig. 3c, the decrease of NPRL2 under siVWCE is convincing, but the decrease of NPRL3 is subtle. How about the other component DEPDC5? It is not shown in the figure. Also, in Fig. 3d-e, only NPRL2 decrease is shown. Is siVWCE affecting specifically NPRL2 or the whole GATOR1 complex?

We thank the reviewer for the constructive comments. We have followed the reviewer’s advice to examine the localization of DEPDC5 utilizing both Lyso-IP and immunostaining assays (Figs. 3d, g, h). Our findings indicate that knockdown of VWCE also affects the lysosomal localization of DEPDC5, implying that the VWCE knockdown impacts the lysosomal localization of the entire GATOR1 complex.

The authors state that “VWCE may serve as a potential therapeutic target for the treatment of prostate cancer”. Besides the *in vitro* colony formation assay, xenografting of prostate cancer cells overexpressing VWCE onto nude mice may better support this point.

We have followed the reviewer’s advice to carry out the xenograft assays (Figs. 4j-l and Extended Data Figs. 7e-j). Consistent with the *in vitro* colony formation assay, overexpression of VWCE suppressed the tumor growth of prostate cancer cells in nude mice.

The authors should discuss why VWCE exert different effects on prostate cancer and liver cancer, and how it may shed light onto the molecular functions of VWCE.

We are grateful to the reviewer for the advice. In the revised manuscript, we have added several sentences to discuss it. The expression level of VWCE is found to be lower in prostate cancer tumors than in the adjacent normal tissues. However, in liver cancers, the expression level of

VWCE remains high in both tumors and normal tissues. It's plausible that the expression of other oncogenic factors may counteract the tumor-suppressing effect of VWCE in liver cancers.

In an effort to further understand how liver cancers might overcome the tumor-suppressive effect of high VWCE expression levels, we conducted the following analysis. First, we identified 17 transcription factors (TFs) co-expressing with VWCE in both prostate and liver cancers, based on the TCGA database (please see below, Fig. 1a). We then analyzed the expression levels of these 17 TFs in five selected cancer cell lines using the CCLE database (please see below, Fig. 1b). We selected seven TFs (HNF4A, GATA4, ARID3A, CEBPA, VTN, NR1H4, and HHEX), whose expression levels showed the highest correlation with VWCE, for promoter binding analysis. CHIP-seq data from the Encode project demonstrated strong binding of HNF4A, GATA4, and ARID3A to the VWCE promoter region (please see below, Fig. 1c). Knockdown of these three TFs resulted in downregulated VWCE expression, suggesting that HNF4A, GATA4, and ARID3A are potential VWCE TFs in liver cancer cells (please see below, Fig. 1d, e). However, knockdown of these three TFs inhibited the growth of multiple liver cancer cell lines, including HepG2 and Huh7, according to the DepMap database (please see below, Fig. 1f). The oncogenic phenotype of HNF4A, GATA4, and ARID3A might be driven by other oncogenes under the influence of these three TFs. In such a scenario, VWCE might merely be a passenger whose suppressive effect is overwhelmed in liver cancers. This may explain why the overexpression of VWCE did not inhibit the growth of liver cancers.

We believe the aforementioned analysis falls beyond the scope of the current manuscript. Therefore, we have only provided these results in the point-by-point response and have briefly discussed them in the revised manuscript. If the reviewer feels it is necessary to include these data within the main manuscript, we are more than willing to make those adjustments.

Figure 1. a) Co-expression of transcriptional factors with VWCE in liver and prostate cancers. An analysis of genes co-expressed with VWCE in PRAD and LIHC was carried out using the TCGA database. A total of 17 transcription factors (TFs) with a correlation greater than 0.4 are indicated in red. **b)** The expression heatmap of VWCE and the TFs in five selected prostate or liver cancer cell lines, based on the CCLE dataset. **c)** HNF4A, GATA4, and ARID3A bind to the VWCE promoter region. The binding of these TFs to the VWCE promoter region was analyzed using ChIP-seq data from the Encode project. **d)** Knockdown of HNF4A, GATA4, or ARID3A results in the downregulation of VWCE mRNA and protein levels in liver cancer cells. HepG2 and Huh7 cells stably expressing the indicated shRNAs were subjected to RT-qPCR (N=3) or immunoblotting assays. **e)** The knockdown efficiency of HNF4A, GATA4, or ARID3A was measured by RT-qPCR (N=3). **f)** According to the DepMap database, knockdown of HNF4A, GATA4, or ARID3A suppresses the growth of several liver cancer cell lines. Data are mean \pm s.d. *P<0.05; **P<0.01; ***P<0.001; ****P<0.0001 (unpaired Student's *t*-test).

Other minor points:

Fig. 2g. Why is LAMP2 expression so low in the whole cell lysates?

We thank the reviewer for raising this question. The seemingly low expression of LAMP2 is due to the short exposure time during that specific immunoblotting experiment. We have repeated the LAMP2 immunoblotting using the same samples and extended the exposure time, as shown in Fig. 2g.

Fig. 2h. Immunoblotting of SZT2 needs to be shown to confirm its efficient knockdown.

We appreciate the reviewer's suggestion. However, an anti-SZT2 antibody is not commercially accessible. The knockdown efficiency of SZT2 was validated by a decrease in the SZT2 mRNA level as determined by the qPCR experiment (Extended Data Fig. 2k) and constitutive activation of mTORC1 during amino acid starvation (Extended Data Fig. 2l).

Fig. 4g. Suppression of mTORC1 activity by VWCE overexpression in the DU145 cell line is not obvious. Similarly, in Fig. 4i, mTORC1 lysosomal localization was not altered significantly in DU145 cells.

We thank the reviewer for raising this question. The DU145 cell line exhibits lower sensitivity to amino acid availability. In the revised manuscript, we have prolonged the amino acid starvation period to 2 hours and followed it with a 15-minute amino acid restimulation, as mentioned in the legend of Extended Data Fig. 6. We found that overexpression of VWCE also inhibited the activation and lysosomal localization of mTORC1 in DU145 cells (Extended Data Figs. 6a, b, h, l). The immunoblotting assays were quantified and compiled with data from three independent replicates.

REVIEWER COMMENTS

Reviewer #1 (Remarks to the Author):

The authors have added new data to support their conclusion. The authors have thoroughly addressed this reviewer's comments.

Reviewer #2 (Remarks to the Author):

This is a resubmission of paper that I previously reviewed. While the authors have made several improvements based on comments of the three reviewers, some aspects of the paper I remain dissatisfied with. For example, a significant concern I voiced in my original review was about the plausibility that VWCE, a protein with all the hallmarks of a secreted protein, is localized to the cytoplasmic face of lysosomes. Early in the manuscript, when the authors first introduce VWCE and propose to investigate its role in mTORC1 function, they need to pre-empt our scepticism by describing the molecular features of VWCE and explaining why they nonetheless believe a role for this protein in mTORC1 biology is plausible. I appreciate the effort that the authors have made in addressing my concern and performing protease treatment of lyso-IP'd lysosomes as I suggested. However, given that they went to the considerable effort to FLAG-tag the endogenous VWCE gene, I really wish they had performed this assay with their 'endogenous' VWCE-expressing cells rather than using cells with 'exogenous' stably-expressed VWCE. I think it remains a distinct possibility that FLAG-tagged VWCE expressed by an exogenous promoter may not behave exactly like the endogenous protein.

On that note, an important concern raised by another reviewer was the relative importance of endogenous VWCE more generally to the proposed model. I think this remains a valid concern and the treatment of this in the manuscript could be improved. For example, I am confused by Extended Data Figure 2a as it is presented. If this is a 'knock-in' experiment, where the endogenous VWCE gene has been FLAG-tagged, then this needs to be more explicitly explained in the manuscript text or Figure legend, etc. Furthermore, how do the authors know that the red arrow is pointing to VWCE-FLAG? What are the extremely prominent bands below the alleged VWCE-FLAG band? It would have been nice to see an anti-FLAG Western blot of whole cell lysates (not immunoprecipitated samples) for both 'WT' and 'knocked-in' cells (with molecular weight markers included) to convince readers that this red arrow is in fact pointing at VWCE. Furthermore, demonstrating that this band is also sensitive to VWCE siRNA would improve our confidence.

Overall, while I don't doubt that VWCE may be acting as a tumour suppressor by some mechanism, and that this mechanism might involve mTORC1 regulation through KICSTOR-GATOR1, I am left with the impression that the findings remain somewhat preliminary.

Reviewer #3 (Remarks to the Author):

The authors have added western quantifications and mouse xenograft experiments to strengthen the quality of the data. My concerns have been satisfactorily addressed. But I do have one more suggestion:

The authors added IF images showing VWCE cellular localization in Extended Data Fig. 2h. This was done by antibody staining in HEK293T cells overexpressing VWCE-FLAG. The quality of the image is not ideal for convincingly resolving the subcellular localization, as the red color seems to be everywhere. Since the authors have generated the VWCE-FLAG-P2A-EGFP knock-in cells, why not check where the endogenous VWCE is expressed in those cells by IF?

Reviewers' comments:

Reviewer #1:

The authors have added new data to support their conclusion. The authors have thoroughly addressed this reviewer's comments.

Reviewer #2:

This is a resubmission of paper that I previously reviewed. While the authors have made several improvements based on comments of the three reviewers, some aspects of the paper I remain dissatisfied with. For example, a significant concern I voiced in my original review was about the plausibility that VWCE, a protein with all the hallmarks of a secreted protein, is localized to the cytoplasmic face of lysosomes. Early in the manuscript, when the authors first introduce VWCE and propose to investigate its role in mTORC1 function, they need to pre-empt our scepticism by describing the molecular features of VWCE and explaining why they nonetheless believe a role for this protein in mTORC1 biology is plausible. I appreciate the effort that the authors have made in addressing my concern and performing protease treatment of lyso-IP'd lysosomes as I suggested. However, given that they went to the considerable effort to FLAG-tag the endogenous VWCE gene, I really wish they had performed this assay with their 'endogenous' VWCE-expressing cells rather than using cells with 'exogenous' stably-expressed VWCE. I think it remains a distinct possibility that FLAG-tagged VWCE expressed by an exogenous promoter may not behave exactly like the endogenous protein.

We thank the reviewer for taking the time to re-evaluate our manuscript and for providing detailed and insightful feedback that has enhanced our work. We have followed the reviewer's advice to provide a more comprehensive description of the molecular features of VWCE early in our manuscript (please see the second paragraph of the main text).

We have also followed the reviewer's advice to conduct lyso-IP experiments combined with trypsin digestion using cells expressing endogenous FLAG-tagged VWCE (Extended Data Figs. 2j, l in the revised manuscript). Our findings indicate that a portion of the VWCE proteins is located on the cytosolic surface of the lysosomes.

On that note, an important concern raised by another reviewer was the relative importance of endogenous VWCE more generally to the proposed model. I think this remains a valid concern and the treatment of this in the manuscript could be improved. For example, I am confused by Extended Data Figure 2a as it is presented. If this is a 'knock-in' experiment, where the endogenous VWCE gene has been FLAG-tagged, then this needs to be more explicitly explained in the manuscript text or Figure legend, etc. Furthermore, how do the authors know that the red arrow is pointing to VWCE-FLAG? What are the extremely prominent bands below the alleged VWCE-FLAG band? It would have been nice to see an anti-FLAG Western blot of whole cell lysates (not immunoprecipitated samples) for both 'WT' and 'knocked-in' cells (with molecular weight markers included) to convince readers that this red arrow is in fact pointing at VWCE.

Furthermore, demonstrating that this band is also sensitive to VWCE siRNA would improve our confidence.

We apologize for any ambiguity caused by our presentation of Extended Data Figure 2a. This figure indeed depicts a 'knock-in' experiment in which the endogenous VWCE gene is FLAG-tagged at its C-terminus. We've made modifications to the "Generation of the knock-in cells" section in our methods to articulate this more clearly and have also updated the figure legend for better clarity.

We have incorporated the reviewer's suggestions and added molecular weight markers to Extended Data Figure 2a. Notably, the extremely prominent bands below the alleged VWCE-FLAG band are non-specific bands that only appear during anti-FLAG immunoprecipitation. To more comprehensively address this issue, we have followed the reviewer's advice to carry out anti-FLAG immunoblotting using whole cell lysates from both 'WT' and 'FLAG-VWCE knocked-in' cells (with molecular weight markers included). Our findings confirm that the VWCE band is sensitive to VWCE siRNA (Extended Data Fig. 1f of the revised manuscript).

Overall, while I don't doubt that VWCE may be acting as a tumour suppressor by some mechanism, and that this mechanism might involve mTORC1 regulation through KICSTOR-GATOR1, I am left with the impression that the findings remain somewhat preliminary.

We thank the reviewer for this comment. Our primary goal with this manuscript was to introduce and provide initial evidence for a novel mechanism by which VWCE might exert its function in mTORC1 regulation. We are confident in the initial findings we have presented, but we also recognize the need for more in-depth studies to elucidate the broader impact of this mechanism in the context of tumorigenesis. It is pivotal to our ongoing research to discern how VWCE influences the KICSTOR complex, as this understanding is key to further establish the KICSTOR-GATOR1-dependent role of VWCE in tumor suppression. We value the reviewer's insightful remarks, which will certainly guide our future research endeavors in this area.

Reviewer #3:

The authors have added western quantifications and mouse xenograft experiments to strengthen the quality of the data. My concerns have been satisfactorily addressed. But I do have one more suggestion:

The authors added IF images showing VWCE cellular localization in Extended Data Fig. 2h. This was done by antibody staining in HEK293T cells overexpressing VWCE-FLAG. The quality of the image is not ideal for convincingly resolving the subcellular localization, as the red color seems to be everywhere. Since the authors have generated the VWCE-FLAG-P2A-EGFP knock-in cells, why not check where the endogenous VWCE is expressed in those cells by IF?

We appreciate the reviewer's positive feedback on our work. We have followed the reviewer's advice to investigate the endogenous VWCE localization using VWCE-FLAG-P2A-EGFP knock-in cells. Our findings reveal that the presence of VWCE on lysosomes remains unaffected by the

amino acid status (ED Figs. 2h, i of the revised manuscript). In addition, we performed lyso-IP experiments utilizing endogenous FLAG-tagged VWCE cells. These data also suggest that a portion of the VWCE proteins resides on the lysosomes (ED Figs. 2j, l of the revised manuscript).

REVIEWERS' COMMENTS

Reviewer #2 (Remarks to the Author):

My concerns have been adequately addressed and I support publication of this manuscript.

Reviewer #3 (Remarks to the Author):

My concerns have been addressed.

Responses to reviewers:

Reviewer #2 (Remarks to the Author):

My concerns have been adequately addressed and I support publication of this manuscript.

Reviewer #3 (Remarks to the Author):

My concerns have been addressed.

We sincerely thank the reviewers for their positive feedback and support of our manuscript.